# Natural Disaster Tweets Classification Using Multimodal Data

**Mohammad Abdul Basit[1], Salman Ghufran Shaikh[2], Bashir Alam[1]** and **Zubaida Fatima[3]**

[1]Department of Computer Engineering, Jamia Millia Islamia, India

[2]King Abdullah University of Science and Technology, Saudi Arabia

[3]Department of Electronics and Communication Engineering, IIIT-Delhi, India

mohammadbasit3@gmail.com   salman.shaikh@kaust.edu.sa

balam2@jmi.ac.in   zubaidafh@gmail.com

## Abstract

Social media platforms are extensively used for expressing opinions or conveying information. The information available on such platforms can be used for various humanitarian and disaster-related tasks as distributing messages in different formats through social media is quick and easy. Often this useful information during disaster events goes to waste as efficient systems don't exist which can turn these unstructured data into meaningful format which can ultimately assist aid agencies. In disaster identification and assessment, information available is naturally multimodal, however, most existing work has been solely focused on single modalities e.g. images or texts separately. When information from different modalities are integrated, it produces significantly better results. In this paper, we have explored different models which can lead to the development of a system that deals with multimodal datasets and can perform sequential hierarchical classification. Specifically, we aim to find the damage and its severity along with classifying the data into humanitarian categories. The different stages in the hierarchical classification have had their respective models selected by researching with many different modality specific models and approaches of multimodal classification including multi task learning. The hierarchical model can give results at different abstraction levels according to the use cases. Through extensive quantitative and qualitative analysis, we show how our system is effective in classifying the multimodal tweets along with an excellent computational efficiency and assessment performance. With the help of our approach, we aim to support disaster management through identification of situations involving humanitarian tragedies and aid in assessing the severity and type of damage.

## 1 Introduction

Natural disasters and calamities have long been occurring. The gravity of these disasters is dependent upon a number of factors, these could be lives lost, damage caused, economic loss, etc. Each year, these disasters take place in the form of earthquakes, floods, tornadoes and severe storms, hurricanes and tropical storms, droughts and wildfires and cause plenty of damage to the human society. The impact due to these disasters cannot be neglected and thus first-responders such as the local residents, health professionals and emergency workers must come to the aid of the damaged sites and affected individuals. The immediate priority that follows after a disaster occurs is providing emergency aid to the injured people. However, it is also very important to assess the damage that has been caused to different structures and zones as that is a good indication of the severity of the situation.

In times of disaster or a calamity, the information available on social media could be widely used in disaster assessment, management and analysis and for other humanitarian tasks. The data regarding these disasters that are collected using field surveys are often not available immediately. Disaster damage data such as the location, area or the zone and extent of the damaged structures and facilities is critical in disaster management operations (Hao and Wang, 2020). Therefore, it becomes necessary for the first-responders of humanitarian aid to rely on data derived from social media, as many users post messages and information in different formats, i.e image, text, audio, etc. However, the utilisation of this data is not easy. One of the challenges that needs to be tackled before the data can be utilised appropriately is related with isolation of content useful for crisis management. In times of a crisis, it is extremely important to determine which areas have been affected the most (hence require the most attention and should be an immediate priority) and other humanitarian tasks that include missing/found people, injured or dead people and other sub-tasks. However, this becomes difficult to

identify as during these times, there will be a huge volume of data that could mask the severity level of certain areas and people in need. Therefore, there is a need for automatic systems to assess, identify and analyse the damage that has been caused which in turn could benefit the emergency management process notably (Agarwal et al., 2020). Although the extraction of information from social media resources to help humanitarian aid workers has been going on for some time, this has been limited to unimodal data. While these unimodal damage analysis frameworks are efficient, they are not able to assess damage as effectively for social media posts (Agarwal et al., 2020) that come in a multimodal format, whether it be text with images, text with audios, audios with images, etc. Thus, we intend to develop an effective system that can leverage multimodal data and is able to assess, identify and analyse the severity of the damage caused and identify other humanitarian sub-tasks in real time. The unimodal text feature is often not very helpful in disaster situations as many times the text of such tweets is incomplete or incomprehensible and the main idea is often conveyed by the images, leading us to choose the text and image modalities, which can give more information about an event (Ofli et al., 2020). The main reason for using multimodal data sources is that it is possible to extract complementary and richer information coming from multiple sensors, which can provide much more optimistic results than a single input. Some monomodal learning systems have significantly increased their robustness and accuracy, but in many use cases, there are shortcomings in terms of the universality of different feature levels and inaccuracies due to noise and missing concepts (Bayoudh et al., 2022). Even though the approach is beneficial, in practice it is very challenging (Atrey et al., 2010) due to the different noise and conflicts in the different modality data (Wang et al., 2007). Therefore one has to find a model which can balance the same. We explore different models on both the modalities and select the one which best balances all the noise present in the different modalities. Further the design of the system should be such that it is responsive, fast and efficient in its computation as the response is needed quickly during a disaster. The system should also have scope for addition and deletion of different categories which is only possible if the architecture is a modular one and as a result we choose the hierarchical structure. The system

should also be able to handle unimodal tweets (text only). We leverage the recent advances in classification for images and text to build an efficacious system which can give much fine grained results.

## 2 Related Work

In the context of unimodal classification, Madichetty and Sridevi (2019) classified textual tweets from CrisisMMD (Alam et al., 2018) dataset into informative and non-informative categories using CNN and ANN. Alam et al. (2019) develop an automatic data processing service which takes in textual data from various sources and classify it into disaster type, informative and humanitarian information conveyed. For the purpose of classification, deep learning and classical algorithms are used. While these unimodal approaches produce commendable results, they have been surpassed by the multimodal approaches. Hao and Wang (2020) propose a data-driven method to locate and assess disaster damage with humongous multimodal social media data. The images have been classified using machine learning while the text follows a keyword search based method. Gautam et al. (2019) analyse the multimodal data related to different natural disasters, for instance: floods, earthquakes, etc. They propose a novel decision diffusion technique to classify them into informative or non-informative categories. Their approach of training an image and text classifier and combining the two outperforms the baselines. Zou et al. (2021) proposed a method through which they integrated image and text information to identify disaster images collected from different social media platforms. They use a deep learning method and FastText framework to extract visual and textual features respectively. They then develop a data fusion model to combine these features and experiment on real world disaster dataset through the CrisisMMD dataset. Abavisani et al. (2020) present a new multimodal fusion method that uses both images and texts as input. They introduce a cross-attention module that is able to filter out uninformative and misleading components. Agarwal et al. (2020) present a damage identification and its severity detection system, called Crisis-Dias. Through qualitative, quantitative and theoretical analysis on a real-world multimodal dataset, they are able to show that the information once presented together, often produces high-end analysis about the domain and

facilitates better learning performance. Mouzannar et al. (2018) propose a multimodal deep learning framework to identify damage related information. In their approach, they combine multiple pretrained unimodal CNNs that extract features from the raw texts and images respectively. Finally, a resultant classifier labels the posts based on both modalities. Their results on a home-grown database of labelled social media posts shows good results and validates the use of the proposed method. Nalluru et al. (2019) experiment with combination of semantic textual features with the image features to be able to efficiently classify a relevant multimodal social media post. They utilise a feature generate framework which uses TF-IDF vectors and glove embeddings for the textual data, and pre-trained residual networks for generating vectors for the image data, the vectors extracted are then concatenated through which they build a LightGBM model to classify different disaster-events as informative or not. Their results demonstrate that features based on a hybrid framework (using both text and images) improve the performance of identifying relevant posts.

## 3 Dataset and Problem Definition

The dataset that was used for this project was the CrisisMMD[1] dataset. It's labels were aggregated according to the specific task at hand. The dataset included information that could be filtered extensively depending on the scenario. The CrisisMMD dataset consists of thousands of annotated tweets and images collected during seven various natural disasters including earthquakes, hurricanes, wildfires, and floods that happened in the year 2017 across different parts of the world. The data is hierarchical as the class labels at each stage depend on the annotation in the previous stage. The above data is further complemented by a text only dataset called HumAID[2] for certain downstream tasks. We use the several sub-categories of the classes to produce our desired results which is to be able to identify the structure damaged and it's severity along with humanitarian mishaps. To do this, based on the dataset we formulate the following tasks.

---

[1]https://crisisnlp.qcri.org/crisismmd
[2]https://crisisnlp.qcri.org/humaid_dataset

### 3.1 Task 1

The objective of this stage is to distinguish between Informative and Non-Informative samples. Any image or text which contains useful events which points to any of the signs of damage, suffering during a disaster has been labelled as Informative. The dataset has separate labels for the image and text. For all tweets $M(t, i)$ where $t$ is the text and $i$ is the image, we formulate a binary function $F_{informative}$. The data has separate labels for both image and text.
Class Distribution:
*Text: Informative: 11509 - Non Informative: 4549*
*Image: Informative: 9374 - Non Informative: 8708*

### 3.2 Task 2

The CrisisMMD dataset labels the tweets into further sub classes depending on whether they signify structural damage or people being affected in any way. The classes are so chosen that they offer critical information to first respondents.
Knowledge of tweets conveying structural and humanitarian rescue information would be of great assistance to relief workers. To accomplish this, the classes from the original dataset have been aggregated into 3 main classes for the system. The classes *Affected Individuals, Injured or dead people, Missing or found people, Rescue Volunteering or donation effort* have been grouped into *Humanitarian*, the *Infrastructure and Utility Damage, Vehicle Damage* into *Structure* and *Not humanitarian* and *Other relevant Information* into *Non Informative*. The textual tweets have been enriched with tweets from the HumAID (Firoj Alam, 2021) dataset to reduce the class imbalance for the structure and humanitarian classes. The dataset contains text only tweets from natural disasters between 2016-2019 and labelled under similar labels as the CrisisMMD, as a result they increase the diversity of the dataset along with reducing the class imbalance. We formulate a tri-valued function $F_{human-struct}$ for a given tweet $M(i, t)$ where $i$ is the image and $t$ is the text.

The purpose of aggregating the sub classes into more generic classes at this level is so that the number of classes to classify remain small along with events having similar features being clubbed into a single category which can later undergo finer classification into their respective sub classes. We include the Non Informative class at this stage and

the upcoming ones since it is a hierarchical system, any samples earlier miss-classified can be removed at further stages thus reducing the rate of errors. As a result these samples will not be processed any further by the system as they don't contain any useful information. As the above task, the image and text have different labels.

Class Distribution

*Text (After enrichment from HumAID): Humanitarian: 9572 - Non Informative: 9359 - Structure(Damage): 8677*

*Image: Humanitarian: 2917 - Non Informative: 11237 - Structure(Damage): 3928*

### 3.3  Task 3

The tweets from the Structure(Damaged) in the previous task come over to this stage. The objective of this stage is to be able to determine the specific structure which was damaged along with its severity, which would assist the rescue ops to target their efforts in a more targeted and prioritised manner. The tweets which belonged to the structure class have been manually annotated to identify the damaged structure into one of the following classes : *buildings, roads/bridges/vehicles*, and *no structure* and the severity into one of the following : *Severe, Mild* and *No damage*. The structure identification and subsequent severity tagging was done mainly for tweets which contained images with Structure class as the text information is often not enough to estimate the severity of a damaged structure. The *no structure* and *No damage* classes have been included keeping in mind the plausible miss-classifications in earlier stages. The task assigns two labels to every such tweet to indicate the damaged structure and the severity. The distribution of the tweets is as such:

*Structure Damaged : Buildings: 1886 - Roads-bridges-vehicles: 646 - No-Structure: 13547*

*Damage Severity: Severe: 2446 - Mild: 895 - No Damage: 14741*

### 3.4  Task 4

The tweets which are marked as humanitarian in Task 2 make their way to this task. This task aims at further sub classifying the humanitarian tweet to the specific message it conveys. The classes in the original dataset have been combined into more generic classes as there existed multiple classes which conveyed similar information. The classes that we classify into are *people affected*, which is an aggregate of *affected individuals, injured or dead people, missing or found people*, *rescue needed* as *rescue volunteering or donation effort* and all other classes are placed in *no human*. The original dataset has different annotation for image and text but we have assigned a single class for a text-image pair by following the following approach: If either of the tags contain a class from people affected, the entire pair is *people affected*, *rescue needed* follows next and if none are present it falls into *no human*. The *people affected* class has been given more priority over the *rescue needed* as often people injured, in critical conditions need medical care quickly and have to be attended to by paramedics urgently.

Class Distribution:

*People Affected: 1648 - Rescue Needed: 4502 - No Human: 11932*

## 4  Methodology

### 4.1  Text Preprocessing

- **Lower case the text**: We lowercase all the text so that it is in the same case.

- **Remove URLs, Mentions (@), RTs**: We remove all URLs starting with http/https, mentions which start with @ sign, and any RT signs i.e. Retweeted as they are not useful for text classification.

- **Remove unnecessary characters**: Removal of all the extra spaces, operator signs, Non-ASCII characters, punctuation marks and single characters as this solely increases the length of the sample without adding much information.

- **Add space between words and removing extra spaces**: The space has to be inserted between the words whilst making sure there are no extra spaces between two words.

### 4.2  Image Preprocessing

We apply the following augmentation on the images so that the model generalises well over data which it hasn't seen and prevents overfitting of data to the training samples. We perform randomised setting of the following characteristics of the image: flipping, rotation, zoom, height, width. All values are transformed in the range of 0 to 1 along with resizing of the image to the models requirements.

### 4.3 Task 1

The function for this stage is formulated as a simple OR operation.

$$F_{informative}(t,i) = \begin{cases} 1(\text{informative}) & \text{if } \beta_t = 1 \text{ or } \beta_i = 1 \\ 0(\text{non-informative}) & \text{if } \beta_t = 0 \text{ and } \beta_i = 0 \end{cases} \quad (1)$$

where $\beta_x$ represents the output of the model for modality $x$.

The rationale behind using an OR operation is that often images could convey life saving information which the text might miss out and vice versa. For this purpose separate classifiers are trained for both the text and image modalities and the outputs of their classification are combined with an OR operation. The imbalance in textual tweets have been overcome using the SMOTE (Chawla et al., 2002) algorithm. The non informative tweets are dropped and the remaining propagate further down the pipeline.

### 4.4 Task 2

For this stage, we train 2 separate classifiers for the image and the text data in similar fashion to Task 1, after which the results are fused using 3 valued function $F_{human-struct}$ as below.

$$F_{human-struct}(t,i) = \begin{cases} 2(\text{Structure}) & \text{if } \beta_t = 2 \text{ or } \beta_i = 2 \\ 0(\text{Humanitarian}) & \text{elif } \beta_t = 0 \text{ or } \beta_i = 0 \ (2) \\ 1(\text{Not Informative}) & \text{else} \end{cases}$$

where $\beta_x$ represents the output of the model for modality $x$.

The structure has been given more priority as it is imperative to figure out major structural damages which could otherwise obstruct relief operations. For text the imbalance is resolved using the HumAID dataset while for images after trying a variety of methods we have stuck with class weights which give more weightage to samples from minority classes.

### 4.5 Task 3

This deals with the structure/damage related tweets from stage 2. In this, we consider 2 approaches. In the first we obtain a common feature vector for both the text and image modalities and use it to train a multi task classifier to predict damaged structure and severity. In the second approach we still use the combined feature vector but to only predict the damage severity, the structure damaged is classified using a separate image classifier i.e. we train 2 separate classifiers. Since we are using multiple feature vectors from multiple modalities we need to have a common representation of the both

so that the individual noise present in the different modalities doesn't degrade the performance of the classifier. For this we are using the Projection layer. To handle class imbalance in the data we use class weights and sample weights by assigning more importance to minority class samples. The reason we are experimenting with multi task learning is because the two tasks are similar, they can share the same representation and there is common information between the two.

### 4.6 Task 4

The tweets which are marked as humanitarian in Task 2 make their way to this task. This task follows a similar approach as the second approach in Task 3, i.e. to train a classifier by combining the image and text feature vectors into one. The class imbalance is handled by class weights.

## 5 Experimental Setup

We experiment with various state of the art text embeddings and image embeddings utilising newer models like Image transformers in addition to already successful CNN based models. The models have been fine tuned and while some of them have had partial training to fit our use-case. The models are run for a maximum of 100 epochs but with early stopping and a set patience value. The models with best validation loss and recall are stored at every stage. The optimiser used in the respective models vary between Adam (Kingma and Ba, 2017) , AMSGrad (Reddi et al., 2018) and Stochastic Gradient Descent. The activation functions are either ReLU or Softmax. The Cross Entropy loss function is used which can be binary or categorical based on the number of classes. The tokeniser used for Bert and its variants are BERT tokeniser[3], GPT tokeniser[4] for GPT2 and its variants and the Keras tokeniser[5] for others. The projection layer in Task 3 takes as input the embedding produced by an encoder and applies the Gaussian Error Linear Unit (Hendrycks and Gimpel, 2016) activation function on it. It then passes it through a dense layer of 256 neurons. The multiple vectors are then added using the Add layer and a single vector is obtained. The output passes through a layer

---

[3] https://huggingface.co/docs/transformers/model_doc/bert
[4] https://huggingface.co/docs/transformers/model_doc/gpt2
[5] https://keras.io/api/keras_nlp/tokenizers/tokenizer/

| Task | Model | Precision | Recall | F1-Score | Acc. |
|------|-------|-----------|--------|----------|------|
| Task1 Text | Word2Vec | 0.80 | 0.78 | 0.79 | 0.83 |
| | **BERT** | **0.81** | **0.79** | **0.80** | **0.83** |
| | FastText | 0.82 | 0.77 | 0.79 | 0.83 |
| | Glove | 0.79 | 0.78 | 0.79 | 0.82 |
| | XLNet | 0.81 | 0.76 | 0.78 | 0.83 |
| Task1 Image | DeIT | 0.81 | 0.81 | 0.81 | 0.81 |
| | DenseNet | 0.79 | 0.79 | 0.79 | 0.79 |
| | Xception | 0.82 | 0.82 | 0.82 | 0.82 |
| | **ResNet** | **0.82** | **0.81** | **0.81** | **0.81** |
| | Inception V3 | 0.81 | 0.80 | 0.80 | 0.80 |
| **Multi-modal** | **BERT-ResNet** | **0.83** | **0.77** | **0.79** | **0.86** |
| Task2 Text | **ALBERT** | **0.77** | **0.81** | **0.79** | **0.85** |
| | BERT | 0.75 | 0.80 | 0.77 | 0.84 |
| | DistilBERT | 0.74 | 0.79 | 0.76 | 0.83 |
| | DistilGPT-2 | 0.74 | 0.80 | 0.76 | 0.83 |
| | XLNet | 0.73 | 0.80 | 0.76 | 0.82 |
| Task2 Image | Xception | 0.70 | 0.76 | 0.72 | 0.75 |
| | ResNet | 0.73 | 0.78 | 0.75 | 0.79 |
| | Inception V3 | 0.77 | 0.80 | 0.78 | 0.83 |
| | BiT | 0.76 | 0.80 | 0.78 | 0.82 |
| | CaIT | 0.76 | 0.81 | 0.78 | 0.82 |
| | **DeIT** | **0.80** | **0.82** | **0.81** | **0.85** |
| **Multi-modal** | **ALBERT-DeIT** | **0.80** | **0.82** | **0.80** | **0.81** |

Table 1: Results of Task 1 and Task 2

normalisation and the vector can be used as a feature vector. The models and their F1 score, Recall, Accuracy and Precision are reported and the best model is selected based on the accuracy and recall for the informative class. The learning rates for the models are mostly 0.001 except for certain cases where it is 0.01 or 1e-5. The experiments have been performed on a NVIDIA TESLA P100 GPU.

| Task | Model | Class | Precision | Recall | F1-Score | Acc. |
|------|-------|-------|-----------|--------|----------|------|
| Task3 | Multitask ResNet-BERT | Damage | 0.83 | 0.88 | 0.86 | 0.92 |
| | | Structure | 0.73 | 0.85 | 0.78 | 0.91 |
| | **Multitask DeIT-ALBERT** | **Damage** | **0.84** | **0.88** | **0.86** | **0.92** |
| | | **Structure** | **0.80** | **0.88** | **0.83** | **0.93** |
| | BERT-Inception | Damage | 0.78 | 0.81 | 0.79 | 0.90 |
| | BERT-DeIT | Damage | 0.80 | 0.85 | 0.82 | 0.90 |
| | VGG16 | Structure | 0.71 | 0.83 | 0.76 | 0.89 |
| | ResNet152 | Structure | 0.79 | 0.84 | 0.81 | 0.92 |
| Task4 | **Inception-BERT** | **Humanitarian** | **0.88** | **0.84** | **0.86** | **0.89** |
| | DeIT-BERT | Humanitarian | 0.85 | 0.84 | 0.84 | 0.88 |

Table 2: Results of Task 3 and Task 4

# 6  Results

For Task 1, we have selected BERT (Devlin et al., 2018) for the text pipeline as it outperforms all the others. For Image we can clearly see that Xception (Chollet, 2017) performs better than all the models followed by ResNet (He et al., 2015). But despite the higher results we will not choose Xception (77%) as ResNet (80%) has a higher recall for the informative class and our main aim is to reduce the false negatives (we don't want the informative tweets to be labelled as non informative). Upon performing an OR operation on the results of the selected image and text pipelines we see that the accuracy after using data from both modalities has increased to 86%. Which is higher than any of the accuracy of the single modalities. Also the recall of the informative class has also increased to 94%, again more than both the unimodal recalls. ALBERT (Lan et al., 2019) has the higher values for all the metrics and since it is light weight we will be choosing that for our text classification in Task 2. It also has good recall for the classes that are considered the positive ones. For Image, We see that the DeIT (Touvron et al., 2021) has obtained higher recall and Precision than all the other models. As a result we also see how the vision transformers are better at image classification task. We see that after combination of the prediction of modalities, the accuracy has not increased but the quality of the predictions has. The structure class had pretty low metrics for ALBERT (recall: 72%) and the DeIT model had low recall for the humanitarian class (72%), but upon combination there is a balancing effect and the results of all the 3 classes have had a boost. Therefore it is somewhat of an ensemble effect. The recall of the damage class has increased as a result of adding the Image model and the humanitarian class has seen an increase in the recall due to the addition of the text model. After testing the 2 approaches in Task 3, we concur that MultiTask DeIT-ALBERT has better metrics. While naturally the single output models should have had better results, the deviation is due to the noise which could be present in a certain target output in the training samples as a result the model could have been biased towards something. Where as in the Multimodal the model has to concurrently optimise the loss function of the two outputs as a result it can cancel the noise which was present in one target specific sample and absent in the other. For Task 4, the performance of Inception and DeIT

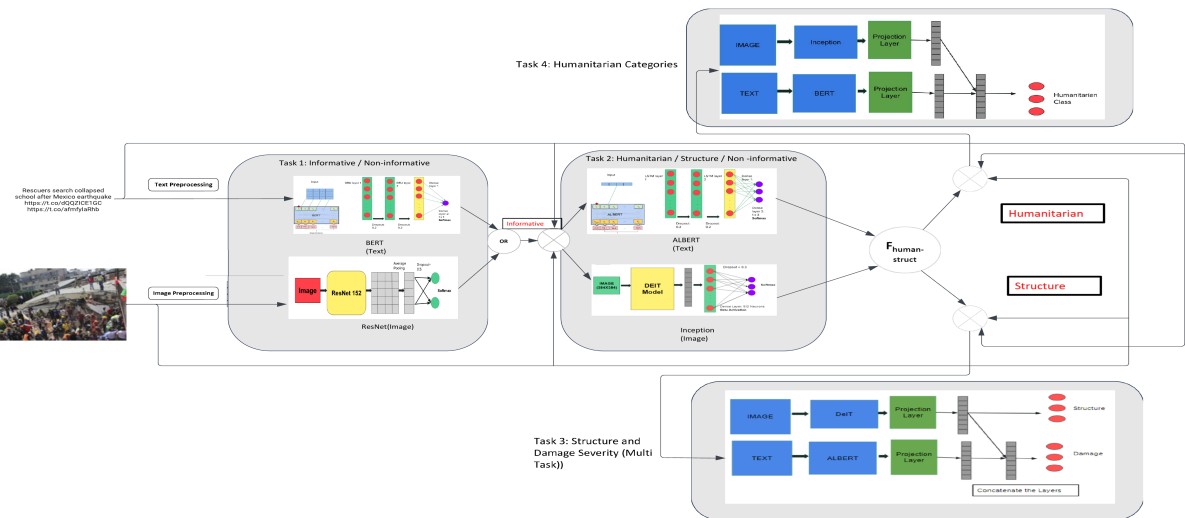

Figure 1: Final System Diagram

is almost comparable but the DeIT has a higher computation cost, for this specific task the DeIT took 8 times the time taken by Inception to do the same task. Since time is of the essence in our application, we have chosen the Inception model (Szegedy et al., 2016) for this task.

## 7  Final System Discussion

Based on our experiments we have selected the components as discussed above and the detailed final system is shown in Figure 1.

**Design:** The embeddings from BERT used for text in the Task 1 are a summation of the last four layers, which is then input to GRU layer. The ResNet 152 model in the image pipeline has all its layers unfrozen except the batch normalisation layers, which are kept frozen so that the learnings of the ImageNet dataset are preserved. The informative tweets progress to Task 2 and the others are dropped at this stage. At stage 2, the text feature vector is obtained by concatenating the output of the last 3 layers of the ALBERT model, which is then fed to the LSTM layer. The DeIT used for Image is a pretrained model which is used to encode the images and then classified using an ANN of size 512 neurons. The outputs are fused using the $F_{human-struct}$. The Structure/Damage related tweets reach the Task 3. In Task 3 we use a Multi Task model where the text features are obtained from ALBERT and DeIT is used for the Images. The reason for taking only the image projections for structure identification is that the structure is usually conveyed from the

image and inclusion of text will not help in most samples. The weightage given to the text and image embeddings are same and both contribute equally in the prediction of the damage severity. In Task 4, the text is fed to the BERT encoder and image to Inception, the output of these 2 encoders are concatenated to form the feature vector. The models used in Task 4 are pretrained.

**Performance:** To analyse the effectiveness of the system, we calculate the F1-score and accuracy of the system as the average of the 4 tasks, the F1-score and accuracy comes out to be 83% and 88% respectively, which is quite impressive. On top of having an accurate classification, the system also has to be responsive and provide quick predictions as it will be used in disaster sites which are time critical. The system is tested on a GPU as well as an older machine on a CPU to see how well it performs in constrained environments. To estimate the time that our model takes in inference we run around 100 tweets and calculate the time a tweet takes to be processed per Task. The inference was run separately on a Intel Core i7 6th generation CPU with 2.6GHz speed (an old CPU) and NVIDIA Tesla P100 (GPU).

$$T_{total} = T_{task1} + T_{task2} + max(T_{task3}, T_{task4}) \quad (3)$$

We find on average a tweet takes 400 milliseconds on the GPU and 1.32 seconds on the CPU over the entire process. The model boasts an impressive speed on a GPU as well as decent processing speeds in the absence of complex architectures as evident from the old CPU's processing time, thus showcasing the resilience of the system.

| Task | Model | Precision | Recall | Accuracy | F1-Score |
|---|---|---|---|---|---|
| | Multi Modal Logistic Regression Decision Policy (Gautam et al., 2019) | - | - | 80 | - |
| Task 1 | Unimodal Text Pipleine (Madichetty and Sridevi, 2019) | 76 | 76 | 75.9 | 76 |
| | **Multi-modal BERT-ResNet (Ours)** | **83** | **77** | **86** | **79** |
| Task 3 (Damage Class) | SES-Cross-BERT-DenseNet (Abavisani et al., 2020) | - | - | 72.65 | 59.76 |
| | **Multi Task DeIT-ALBERT (Ours)** | **84** | **88** | **92** | **86** |
| | MultiModal CNN-VGG16 (Ofli et al., 2020) | 78.5 | 78 | 78.4 | 78.3 |
| Task 4 | SES-Cross-BERT-DenseNet (Abavisani et al., 2020) | - | - | **91.14** | 68.41 |
| | **Inception-BERT (Ours)** | **88** | **84** | 89 | **86** |

Table 3: Comparison with similar studies

**Features:** Even if non informative tweets enter a stage, due to the hierarchical nature they can be removed as every stage involves a non-informative class. The data can be viewed at any granularity level by only allowing tweets to go to a particular stage. Further more if newer and more accurate models are present for the tasks at any stage they can be easily incorporated in the system owing to its modular design, giving it a plug and play feature. The variety of fine grained classes in the system allow the rescue workers to formulate a priority list based on a combination of classes and respond to the emergencies in a swift manner. Along with having a great performance for multimodal tweets, the system can generalise well to unimodal tweets as well, as the first 2 tasks are a simple fusion and the absence of a modality doesn't interfere with the others' classification. For the last 2 phases, the vector of the missing modality can be left blank, as a result we can adapt to a variety of tweets without considerable decrease in quality.

**Comparison:** While comparing with similar work in this field, there doesnt exist exact experiments on the classes or the sub divisions which we divide our data into. Though there does exist papers which deal with a subset of our tasks and we can do somewhat of a task by task comparision to evaluate our system with existing state of the art. It can be observed from the results present in Table 3 that our system outperforms models from similar studies for certain subtasks and produces fine grained classifications with greater flexibility by adopting a modular structure.

**Qualitative analysis:** We perform a qualitative analysis of a few images to show how the model predictions work. This also justifies the design choices which we have made in terms of the model selection.

- **Example-1**:

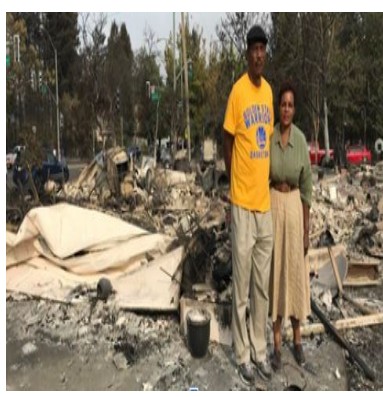

Figure 2: Tweet Text: *California wildfire evacuees just want to "go home" if they have a home still standing https://t.co/wOxKyYDHxt https://t.co/h5RQDtVsHw*

Actual tag: Task 1: *informative*; Task 2: *human*; Task 3: *affected individual*
Predicted: Task 1: Text: *informative* – Image: *informative*; Task 2: Text: *non-informative* – Image: *human*; Task 3: *affected individual*

What we see in Figure 2 is that the classifier predicts all the labels for the image accurately. Even though the text was non informative in the second task, the overall sample is classified as human class due to the $F_{human-struct}$ operation of the predictions of the individual modalities.

- **Example-2**:

It is observed in Figure 3 that the model correctly predicts the tag and the benefit of the 2 modalities is again highlighted when the text and image give their different predictions in Task 2 but we have been able to preserve the important content present in the photo.

# 8 Conclusion

Through our research we have been able to propose a hierarchical model which classifies the given multimodal tweet into the different categories of either humanitarian or structural damage. A thorough

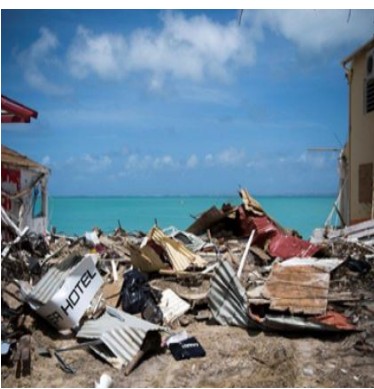

Figure 3: Tweet Text: *The Hotel Heroes of Hurricane Irma https://t.co/Ge3QFiCVte https://t.co/8oA9wbPqeu*

Actual tag: Task 1: *informative*; Task 2: *damage*; Task 3: Severity: *severe* - Structure: *building*
Predicted: Task 1: Text and Image: *informative* ; Task 2: Text- *non informative*; Image- *damage*; Task 3: Severity: *severe* – Structure: *building*

analysis is performed on a combination of Crisis-MMD, HumAID datasets for the purpose of the development of the model. It is also very responsive and can be fitted with different tasks as and when data for such disaster events is available. As a future work on this task, the speed of predictions could be improved gigantically by deploying the models on cloud computing platforms with GPU support and multiple instances with auto scaling. The system could implement zero shot learning as the photos from a natural disaster are not from a closed domain and a model will always not see events similar to what it was trained on. Currently the model has been tested for tweets from Twitter, but similar platforms like Instagram, Facebook, etc. could also be used with this model, we would need to develop an adapter for the particular platform specifying rules for how the image and/or text is obtained from the platform. A future step would be to increase the diversity of the dataset by including samples from myriad social networks. We plan to expand the type of modality to include videos from platforms like YouTube and the like, a precursor to which would be creation of a dataset for the modality. Also the predictions could be further improved by taking user feedback on classifications and using newer vision transformers as well as decision fusion techniques for combining the modalities.

## Limitations

The quality and variety of the data used to train the models varies as no incident can capture all the possible combinations. There was a certain possibility of data being biased and ultimately developing an unfair model. We actively strived to identify and mitigate potential biases in the training data. The system currently works with tweets in English language and further study has to be performed to scale it to more diverse set of languages. The support for unimodal tweets in Task 3 and Task 4 could be improved. Often tweets from humanitarian categories could represent damage categories also, which could be confused by the system. The system depends on the availability of internet connectivity with the masses which could not always be available if the network infrastructure is itself damaged and limits the deployment of the system. Also the system relies on data available on networking sites and expects that the data will always be the truth. Determination of truth-fullness is currently not available.

## Ethics Statement

The intended use of the system is in areas struck with natural disasters and help the greater cause of disaster management. In this project, we used publicly available data, we did not use any data that was considered sensitive or private information that would require explicit consent. We remained committed to stay on top of holding data privacy regulations and respecting individuals privacy rights in terms of data collection. The tweets are represented by tweet IDs which can be used to fetch the actual tweets. If a user restricts or deletes his tweets, it cannot be fetched. Incase of development of a more detailed application which utilises the personally identifiable information, storage guidelines have to be followed. We remained focused on constructive and positive applications, particularly aiding disaster response efforts and explicitly state that the AI model was not used for any illegal activities or harmful purposes.

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
