# OpenReview forum: "Natural Disaster Tweets Classification Using Multimodal Data"
_EMNLP/2023/Conference — EMNLP 2023 Main_

### Official Review · Reviewer_j8SL · 2023-07-26

**Soundness:** 4

**Excitement:**

4: Strong: This paper deepens the understanding of some phenomenon or lowers the barriers to an existing research direction.

**Missing References:**

NA

**Paper Topic And Main Contributions:**

The paper presents a multimodal system that classifies tweets (text and image) into different categories in a hierarchical way. The whole system is organized in separated tasks, four tasks in total with dependency between them.

**Questions For The Authors:**

The authors mentioned, in line 586, they have a reasonable time to execute the whole system, calculating on average 1.32 seconds per tweet. Honestly, the Twitter community produces a lot of tweets per second, and when we dealing with natural disasters quick response is crucial, so in my opinion, 1.32 seconds per tweet is a lot of time. So, be careful with this kind of conclusion or insight.

Second, the authors stated in line 572, they reached an overall f1-score from the whole system of 83%, which I agreed with it seems very impressive, but maybe it would be nice to compare it with other related work, I know that maybe there are no systems similar in the literature, but systems dealing with a subtask of your system yes, so it is important to locate your contribution into the state of the art.
So, I will appreciate some addition to this in the manuscript.

Minor issues:
Figure 1 is very low quality, it is very difficult to appreciate it.
Why used SMOTE in textual tweets and don't in the other kind of task? For instance, in task 4, you used class weights to deal with class imbalance.
Could you give some ideas about why multimodal is not getting better results (with a large margin) versus text only?

**Reasons To Accept:**

The whole system is very interesting to deal with crisis events like natural disasters. Although is not responsibility of the author, the elimination of the API of twitter impossibilities the real performance of this systems.

**Reasons To Reject:**

The authors mentioned, in line 586, they have a reasonable time to execute the whole system, calculating on average 1.32 seconds per tweet. Honestly, the whole community produces a lot of tweets per second, and when we dealing with natural disasters quick response is crucial, so in my opinion, 1.32 seconds per tweet is a lot of time. So, be careful with this kind of conclusion or insight.

Second, the authors stated in line 572, they reached an overall f1-score from the whole system of 83%, which I agreed with it seems very impressive, but maybe it would be nice to compare it with other related work, I know that maybe there are no systems similar in the literature, but systems dealing with a subtask of your system yes, so it is important to locate your contribution into the state of the art.

**Reproducibility:**

3: Could reproduce the results with some difficulty. The settings of parameters are underspecified or subjectively determined; the training/evaluation data are not widely available.

**Reviewer Confidence:**

4: Quite sure. I tried to check the important points carefully. It's unlikely, though conceivable, that I missed something that should affect my ratings.

**Typos Grammar Style And Presentation Improvements:**

The authors have a lot of missing white spaces and also, extra white spaces.
For instance see lines: 17, 135, 352, 569, 453, 419, and more.

---

> ### Author Rebuttal · Authors · 2023-08-27
>
> We appreciate your time and effort in providing us some great and insightful reviews for the paper.
>
> Regarding your questions, we would address them below:
>
> 1. The time it takes to process a single tweet is given for an old CPU only machine to showcase that
> even in circumstances where more complex architectures are not available, the model can operate in
> those situations with a decent processing speed. We do agree that we should have included the time it takes to run on a  GPU as well. The results of running the system on a GPU (Nvidia Tesla P100) is as follows.
>   \\(T_{total} = 400ms\\)
>
>     We clearly see that it takes 1/4th of the time that it did on a CPU only machine. Furthermore when the system is deployed on a production level, it would employ multiple instances and auto scaling measures which would increase  the throughput of the system by bounds. The throughput mentioned above is the time it takes to process one tweet by one instance.
>
>
> 2. The reason we haven't compared with other papers is that, there doesn't exist exact
> experiments on the classes or the sub divisions which we divide our data into. We have introduced a newer category while assessing damage as well as proposed how our output could be used to assign priorities. There are papers which deal with the subset of what we are classifying and any comparison will not be an exact one.
>     But it is definitely a great addition to compare our results with some similar experiments performed by others.
>     We cannot compare the overall F1 score of the system with others as the tasks they perform are
> limited or different in their scope , but we can do somewhat of a task by task comparison.
>
> |      Task      | Model                                                | Precision | Recall | Accuracy  | F1-Score |
> |:--------------:|------------------------------------------------------|-----------|--------|-----------|----------|
> | Task 1         | Multi Modal Logistic  Regression Decision Policy [1] |           |        | 0.80      |          |
> |                | Unimodal Text Pipeline [2]                           | 76        | 76     | 75.9      | 76       |
> |                | **Multi-modal BERT-ResNet (Ours)**                   | **83**    | **77** | **86**    | **79**   |
> | Task 3         | SES-Cross-BERT-DenseNet [3]                          |           |        | 72.65     | 59.76    |
> | (Damage Class) | **Multi Task DeIT-ALBERT (Ours)**                    | **84**    | **88** | **92**    | **86**   |
> | Task 4         | MultiModal CNN-VGG16 [4]                             | 78.5      | 78     | 78.4      | 78.3     |
> |                | SES-Cross-BERT-DenseNet [3]                          |           |        | **91.14** | 68.41    |
> |                | **Inception-BERT (Ours)**                            | **88**    | **84** | 89        | **86**   |
>
> *Comparison with State of the art*
>
> Overall our system with an F1 score of 83% outperforms the models mentioned above and produces much more fine grained classifications offering much more details to the end teams.
>
> 3. We used SMOTE only in Task 1 for texts and not for Images as the they are not very effective when dealing with images as often the images formed could be unrealistic and as a result hinder the learning. Also SMOTE works fine with text data as the feature vectors can be  synthesized based on other minority class and help in the learning whereas in images multiple parameters like spatial information, Visual Interpretation could be altered and result in lower quality data. We stuck with class weights in the remaining tasks as we found the results promising when using them as compared to SMOTE as the class weights force the model to pay more attention to minority class during training by assigning higher penalty for wrong predictions on minority class.
>
> 4. While the accuracy of the multimodal models hasn't increased by a huge margin, overall the quality of the predictions has, as mentioned in the results. For example in Task 2 the text pipeline gave better results for the Humanitarian class while the image pipeline had greater metrics for the Damage class. Upon combination, we saw a balancing effect wherein the Precision and recall for both the classes reached a similar state and a cancelling of the earlier bias towards a class.
>   Nonetheless, The reason for accuracy not improving much could be attributed to the fact that, the image is unrelated to the text, or the textual information presents a greater confidence to a certain class  than offered by image. Also sometimes the image could be blurred or just blank, or in general poor image quality could lead to lesser inference from the image. But in general , as mentioned earlier, We see improvement on a larger picture when utilizing multiple modalities.
>
> We will fix all the redundant spaces in the final version of the paper. Regarding Reproducibility, we have specified the parameters of the core model in the paper itself in the experiments and final system discussion headings along with a mention of setting for all the experiments and relevant code in the supplementary materials. The datasets are publicly available, except the structure annotation performed on a subset of the CrisisMMD data, which can be made available on request.
>
> **References**
>
> *[1] A. Gautam, et al., "Multimodal Analysis of Disaster Tweets," in 2019 IEEE Fifth International Conference on Multimedia Big Data (BigMM), Singapore, Singapore, 2019 pp. 94-103. doi: 10.1109/BigMM.2019.00-38*
>
> *[2] S. Madichetty and M. Sridevi, "Detecting Informative Tweets during Disaster using Deep Neural Networks," 2019 11th International Conference on Communication Systems & Networks (COMSNETS), Bengaluru, India, 2019, pp. 709-713, doi: 10.1109/COMSNETS.2019.8711095.*
>
> *[3] Mahdi Abavisani, Liwei Wu, Shengli Hu, Joel Tetreault, Alejandro Jaimes; Proceedings of the IEEE/CVF Conference on Computer Vision and Pattern Recognition (CVPR), 2020, pp. 14679-14689*
>
> *[4] Ofli, Ferda et al. “Analysis of Social Media Data using Multimodal Deep Learning for Disaster Response.” International Conference on Information Systems for Crisis Response and Management (2020).*

---

### Official Review · Reviewer_SknH · 2023-08-03

**Soundness:** 4

**Excitement:**

4: Strong: This paper deepens the understanding of some phenomenon or lowers the barriers to an existing research direction.

**Paper Topic And Main Contributions:**

This papers deals with approaches based on multimodal datasets in order to propose sequential hierarchical classification. The hierarchical model can give results at different abstraction levels based on dedicated use cases. This paper highlights how the proposed system is effective in classifying multimodal tweets.

**Questions For The Authors:**

- For Task1, did you test a more restrictive combination with informative prediction if βt = 1 AND βi = 1?

- In specific cases (e.g. DeIT), the F-score value could be greater than precision and recall. Could you confirm that is explained by the weighting of classes for F-score calculation?

- The meaning and the use of formula (3) line 583 is not clear. Could you explain this?

- The paper takes into account images and texts from tweets. Do you plan to use other kind of textual data (e.g. news) and other kinds of data (like video) in future work?



**Reasons To Accept:**

- The motivations to exploit social media and multimodality in disaster assessment, management and analysis of the severity of the damage and humanitarian sub-tasks in real time are convincing.

- The four subtasks addressed and methods associated with are clearly summarized.

- Experiments are conducted with several state-of-the-art text and image embeddings.

- Based on the results of experiments a general system (with a clear pipeline summarized in Figure 1) and relevant qualitative analysis are proposed and implemented.

**Reasons To Reject:**

- Weak originality of the approaches used but an attractive combination of up-to-date methods well evaluated and integrated in a final system.

- The 'hierarchical' concept proposed is not really clear. I am not sure that is a right term to use to describe the proposed method. In my opinion, the 'sequential' term could be more relevant.

- The paper should extend future work discussion by considering other kind data.

**Reproducibility:**

3: Could reproduce the results with some difficulty. The settings of parameters are underspecified or subjectively determined; the training/evaluation data are not widely available.

**Reviewer Confidence:**

4: Quite sure. I tried to check the important points carefully. It's unlikely, though conceivable, that I missed something that should affect my ratings.

---

> ### Author Rebuttal · Authors · 2023-08-27
>
> We are glad that you were impressed by our approach to divide the issue into tasks and the problem being solved. Though, to further ameliorate your confidence in our paper, we would like to address your concerns.
>
> We agree with the view that it is indeed a sequential system as a step is preceded by some step, i.e something which is being tested for task 2 has to first be classified as informative by task 1 and so on. But the reason for calling it hierarchical is that their is a sort of tree like pattern in which a certain tweet is classified. For instance an informative tweet can be any of Humanitarian or Structure classes. Furthermore, a structure could be classified into further sub categories based on damage and severity, similarly the humanitarian  class is further sub divided into 3 classes. This hierarchy is what we meant by the title.
>
> **Answers to Questions to Author**
> 1. Yes during the testing of the model for Task 1, we have tested all the four combinations, and specifically for the pair you have mentioned with \\(\\beta_i\\) and \\(\\beta_t\\) as 1 , we had a total of 1688 samples out of which 97% were classified correctly. The combination can work with these samples since it is an OR operation and the accuracy of correct prediction for such samples will be extremely high.
>
> 2. Thanks for your observation, we have investigated the matter, and it is a result of human error. The values for F1 score and accuracy have been swapped for the **Task2 Image** and **Task4**. The issue missed the eyes of the authors despite multiple reviews.
> The change is simple and we would need to swap the values for the 2 columns in the 2 sub categories. This would not have any effect on the final outcome and results (F1 score and Accuracy) as they have been calculated using the correct values for the two and this is just an error on our part while filling the tables.
> After correction the 2 parts of the table 1 and table 2 would look as follows:
>
> | Task     | Model | Precision | Recall | F1-Score | Acc.|
> | :------| :------: | ----: |-------|-------|----|
> |             | Xception   | 0.70 | 0.76 | 0.72 | 0.75 |
> | Task2 |   ResNet   | 0.73 | 0.78 | 0.75 | 0.79 |
> | Image |  Inception  V3 | 0.77 | 0.80 | 0.78 | 0.83 |
> |             |  BiT   | 0.76 | 0.80 | 0.78 | 0.82 |
> |             | CaIT | 0.76 | 0.81 | 0.78 | 0.82 |
> |             | **DeIT**| **0.80** | **0.82** | **0.81** | **0.85** |
>
> *Changes in Table 1*
>
> | Task     | Model |Class| Precision | Recall | F1-Score | Acc.|
> | :------| :------: | ---|----: |-------|-------|----|
> |             | **Inception-BERT**| **Humanitarian** | **0.88*** | **0.84** | **0.86** | **0.89** |
> | Task 4 |   DeIT-BERT | Humanitarian | 0.85 | 0.84 | 0.84 | 0.88 |
>
>  *Mistakenly mentioned as 0.84 in anonymous submission paper
>
>
> *Changes in Table 2*
>
>
> 3. The formula has been used to postulate the time it takes for 1 tweet on average to be predicted by the complete pipeline (go through all stages). The \\(T_{task}\\) specifies the time it takes for a tweet to go through the task. The task 1 and 2 are sequential so their times have been added up while task 3 and 4 could be done in parallel and therefore a max of their value is taken (A tweet can only go for one of the tasks). The time calculated in paper is when the model is run on an older CPU system to show the resilience of the system and how it performs decent even on a non GPU machine. When run on a GPU (ideally a multimodal system should run on a GPU), the model boasts a speed of *400ms/tweet*.
>
>
> 4. Currently we had tested the model primarily for tweets from Twitter (now known as X), but definitely similar platforms like Instagram, Facebook , Weibo , etc could also be used with this model, we would need to develop  an adapter for the particular platform specifying rules for how the image and/or text is obtained from the platform. The adapter could be developed in this sense for any social media platform and it would be good to go with the model. As a future work, a dataset could be curated containing samples from a myriad of social networks and news websites and help in development of a more robust system. The model currently works with textual data, image and a combination of both as already mentioned in paper. Video is currently not supported but we plan on utilizing videos from youtube and similar platforms and integrating that as a modality in the system. A precursor to that would be the curation of a dataset for the same. We have currently not mentioned these in the future discussion but would be open to the edit. Furthermore, in terms of more effective model building, more experimentation could be done with newer vision transformers as well as decision fusion techniques for combining the modalities.
>
> Regarding Reproducibility, we have specified the parameters of the core model in the paper itself in the experiments and final system discussion headings along with a mention of setting for all the experiments and relevant code in the supplementary materials.

---

### Official Review · Reviewer_iDm4 · 2023-08-05

**Soundness:** 4

**Ethical Concerns:**

Yes

**Excitement:**

4: Strong: This paper deepens the understanding of some phenomenon or lowers the barriers to an existing research direction.

**Paper Topic And Main Contributions:**

In this work, the authors present a multi-modal approach to classifying natural disaster tweets. The system incorporates both bert-like transformer models (for text) and deep learning models (for images). The paper is well written and sound, and the authors provide all needed details, code, and data. The evaluation is thorough and the enhanced dataset they produce is a solid contribution on its own. The methodologies used are not new, but the pipeline putting them together and accompanying analysis is a valuable enough contribution.

**Reasons To Accept:**

- Solid augmented dataset for the multi modal task
- nice pipeline for putting together existing tools for a common goal
- marginally improve performance on existing datasets

**Reasons To Reject:**

- Balancing of dataset weakens the contribution. In a real world scenario the imbalance in Twitter data is considerable, rendering the findings not applicable in this scenario.

**Reproducibility:**

5: Could easily reproduce the results.

**Reviewer Confidence:**

4: Quite sure. I tried to check the important points carefully. It's unlikely, though conceivable, that I missed something that should affect my ratings.

---

> ### Author Rebuttal · Authors · 2023-08-27
>
> Thank you for your comments, we were thrilled to see your interest in the paper.
>
> Nonetheless, we would like to clarify on the balancing of dataset. The dataset which we aim to balance is the training data which would be used for the model's initial learning. The balanced dataset helps in achieving greater classification performance and efficient models as has been proven by multiple researches.
> The data which would be fed to the model in a real world scenario would definitely be heavily skewed and that can be handled as the model would only be performing predictions (classification) on the data and wouldn't be used to train it on the go, thus preserving the performance.
>
> Also overtime a balanced data set could be created from the predictions which could be used to further train the model to keep up with the recent trends.

---

### Meta-Review · Area_Chair_NhPp · 2023-09-15

**Recommendation:** 5

**Metareview:**

This paper deals with approaches based on multimodal datasets in order to propose sequential hierarchical classification. The hierarchical model can give results at different abstraction levels based on dedicated use cases. This paper highlights how the proposed system is effective in classifying multimodal tweets. All reviewers appreciated the soundness and excitement of this paper. This paper has great potential for acceptance for presentation at main conference.

Here is a summary of the major points in the reviews.

Reasons To Accept:
- The motivations to exploit social media and multimodality in disaster assessment, management and analysis of the severity of the damage and humanitarian sub-tasks in real time are convincing.
- The proposed system is very interesting to deal with crisis events like natural disasters.
- The new solid augmented dataset for the multi modal task is constructed.
- The nice pipeline for putting together existing tools for a common goal which has the four subtasks is proposed.
- Experiments are conducted with several state-of-the-art text and image embeddings.
- The experiment result shows the proposed multimodal-processing approach marginally improve performance.

Reasons To Reject:
- The used tequniques are not really original (but the paper proposes an attractive combination of up-to-date methods well evaluated)

---

### Decision · Program_Chairs · 2023-10-07

**Decision:**

Accept-Main

**Comment:**

This paper deals with approaches based on multimodal datasets in order to propose sequential hierarchical classification. The hierarchical model can give results at different abstraction levels based on dedicated use cases. This paper highlights how the proposed system is effective in classifying multimodal tweets. All reviewers appreciated the soundness and excitement of this paper. This paper has great potential for acceptance for presentation at main conference.

Here is a summary of the major points in the reviews.

Reasons To Accept:
- The motivations to exploit social media and multimodality in disaster assessment, management and analysis of the severity of the damage and humanitarian sub-tasks in real time are convincing.
- The proposed system is very interesting to deal with crisis events like natural disasters.
- The new solid augmented dataset for the multi modal task is constructed.
- The nice pipeline for putting together existing tools for a common goal which has the four subtasks is proposed.
- Experiments are conducted with several state-of-the-art text and image embeddings.
- The experiment result shows the proposed multimodal-processing approach marginally improve performance.

Reasons To Reject:
- The used tequniques are not really original (but the paper proposes an attractive combination of up-to-date methods well evaluated)